# Fronto-thalamic networks and the left ventral thalamic nuclei play a key role in aphasia after thalamic stroke
Ida Rangus [1,2,14] ✉, Ana Sofia Rios [1,14], Andreas Horn[1,3,4,5,6], Merve Fritsch[7], Ahmed Khalil [1,2], Kersten Villringer[1,2], Birgit Udke[8], Manuela Ihrke [8], Ulrike Grittner [9,10], Ivana Galinovic[1,2], Bassam Al-Fatly [3], Matthias Endres[1,2,10,11,12,13], Anna Kufner [1,2,15] & Christian H. Nolte [1,2,10,11,15]

Thalamic aphasia results from focal thalamic lesions that cause dysfunction of remote but functionally connected cortical areas due to language network perturbation. However, specific local and network-level neural substrates of thalamic aphasia remain incompletely understood. Using lesion symptom mapping, we demonstrate that lesions in the left ventrolateral and ventral anterior thalamic nucleus are most strongly associated with aphasia in general and with impaired semantic and phonemic fluency and complex comprehension in particular. Lesion network mapping (using a normative connectome based on fMRI data from 1000 healthy individuals) reveals a *Thalamic aphasia network* encompassing widespread left-hemispheric cerebral connections, with Broca's area showing the strongest associations, followed by the superior and middle frontal gyri, precentral and paracingulate gyri, and globus pallidus. Our results imply the critical involvement of the left ventrolateral and left ventral anterior thalamic nuclei in engaging left frontal cortical areas, especially Broca's area, during language processing.

Human language is an intricate cognitive system organized within cortico-subcortical networks, which are required to operate in a coordinated fashion to assure undisturbed language production and reception[1,2]. An injury to any element of these language networks could lead to a system malfunction resulting in aphasia.

Besides the classical perisylvian cortical regions of the dominant hemisphere, language strongly depends on subcortical structures, the most prominent one being the thalamus[3–5]. Language impairment resulting from isolated thalamic lesions is termed *thalamic aphasia*[5]. The thalamus is not a homogenous structure, but rather consists of distinct nuclei which are involved in various tasks[6]. So far, the ventral lateral, ventral anterior, dorsomedial and centromedian nuclei, as well as the pulvinar nucleus have been suggested as potential central players in language processing based on observational studies with thalamic lesions of various etiologies such as stroke, tumour, and stereotactic surgery[5,7,8]. However, the exact thalamic site involved in language processing remains elusive, especially with respect to affected language subdomains. Interestingly, while specifics vary, there is consensus that thalamic structures are involved in word fluency tests,

[1]Charité – Universitätsmedizin Berlin, corporate member of Freie Universität Berlin and Humboldt-Universität zu Berlin, Klinik für Neurologie mit Experimenteller Neurologie, Berlin, Germany. [2]Charité - Universitätsmedizin Berlin, corporate member of Freie Universität Berlin and Humboldt-Universität zu Berlin, Center for Stroke Research Berlin (CSB), Berlin, Germany. [3]Charité - Universitätsmedizin Berlin, corporate member of Freie Universität Berlin and Humboldt-Universität zu Berlin, Klinik für Neurologie mit experimenteller Neurologie, Movement Disorder and Neuromodulation Unit, Berlin, Germany. [4]Department of Neurology, Harvard Medical School, Boston, MA, USA. [5]Center for Brain Circuit Therapeutics, Brigham and Women's Hospital, Boston, MA, USA. [6]Department of Neurology, Brigham and Women's Hospital, Boston, MA, USA. [7]Charité - Universitätsmedizin Berlin, corporate member of Freie Universität Berlin and Humboldt-Universität zu Berlin, Klinik für Psychiatrie und Psychotherapie, Berlin, Germany. [8]Charité - Universitätsmedizin Berlin, corporate member of Freie Universität Berlin and Humboldt-Universität zu Berlin, Klinik für Audiologie und Phoniatrie, Berlin, Germany. [9]Charité – Universitätsmedizin Berlin, corporate member of Freie Universität Berlin and Humboldt-Universität zu Berlin, Institut für Biometrie und klinische Epidemiologie, Berlin, Germany. [10]Berlin Institute of Health at Charité – Universitätsmedizin Berlin, Berlin, Germany. [11]German Center for Cardiovascular Research (Deutsches Zentrum für Herz Kreislauferkrankungen, DZHK), Partner Site Berlin, Berlin, Germany. [12]Charité - Universitätsmedizin Berlin, corporate member of Freie Universität Berlin and Humboldt-Universität zu Berlin, NeuroCure Cluster of Excellence, NeuroCure Clinical Research Center (NCRC), Berlin, Germany. [13]German Center for Neurodegenerative Diseases (Deutsches Zentrum für Neurodegenerative Erkrankungen, DZNE), Partner Site Berlin, Berlin, Germany. [14]These authors contributed equally: Ida Rangus, Ana Sofia Rios. [15]These authors jointly supervised this work: Anna Kufner, Christian H. Nolte. ✉ e-mail: ida.rangus@charite.de

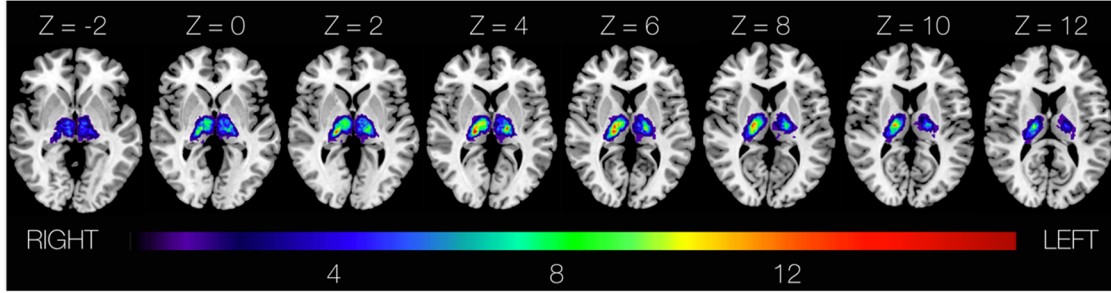

**Fig. 1 | Lesion distribution map.** Overlap of all lesion maps (*n* = 85) superimposed onto an axial view of a template brain ch2better.nii.gz by MRIcron in MNI 152 space[19]. Colour scale indicates the number of patients showing a lesion in a given voxel.

anomia in spontaneous language and confrontation naming, while comprehension and repetition mostly remain well preserved[9–11]. These findings suggest that the thalamus is embedded in language tasks that rely on involvement of the frontal lobe networks.

Apart from mere anatomical localization of the thalamic structures involved in language, an analysis on a network level is essential to fully understand language as a multimodal cognitive function, and aphasia as a network dysfunction. An effective approach to analyse the network-level interactions is lesion network mapping (LNM), which helps identify the relationship between lesion locations causing specific symptoms and their connections to the rest of the brain by using normative resting-state fMRI data from healthy individuals. This method has previously been used to identify networks associated with various conditions such as tremor, tics and addictions[12–15].

Larger, focused, comprehensive studies on the role of thalamic sub-regions and properties of specific thalamo-cortical networks especially with respect to specific language subdomains are still lacking. Ultimately, the identification of well-defined brain networks involved in language tasks affected by thalamic lesions could serve as potential therapeutic targets for interventional therapies of post-stroke aphasia[16,17]. Therefore, we set out to combine two methodologies – lesion symptom mapping (LSM) and LNM – to comprehensively assess thalamic aphasia in a prospective, homogenous cohort of ischaemic stroke patients with unilateral thalamic lesions who received advanced, expert evaluation of language function within seven days of stroke onset.

Our aims were to (i) identify thalamic nuclei primarily associated with worse language impairments and respective impaired language subdomains and (ii) to identify a network of brain regions functionally connected to thalamic lesions that are associated with worse language performance.

In this study, we demonstrated specific involvement of the left ventrolateral and ventral anterior thalamic nuclei in lesions associated with language impairments. Connected to these thalamic nuclei, we observed left-lateralized functional projections involving the inferior frontal gyrus' pars triangularis and pars opercularis in addition to the middle and superior frontal gyri, precentral gyrus, frontal orbital cortex, cingulate (anterior) and paracingulate gyri and basal ganglia (internal pallidum). Our findings suggest that left ventral thalamic nuclei, embedded in fronto-thalamic networks contribute to the multimodal language processing.

## Results
### Demographics
Eighty-five consecutive patients with acute unilateral ischemic thalamic stroke were enroled. The median age was 73 years (IQR 61–81), 39/85 (46%) of patients were female. Stroke severity was mild (median NIHSS on admission = 2 (IQR 1–4)). All patients underwent language assessment via Aphasia Check List (ACL)[18] which gave information on language impairment in general (aphasia score) and affected language subdomains. Language assessment took place very early after symptom onset: median time from symptom onset to MRI = 2 days (IQR 1–2 days) and median time from symptom onset to language assessment = 2 days (IQR 2–3 days).

The average lesion size was 0.50 cm³ (IQR 0.26–0.98 cm³). Lesions were right-sided in 39 and left-sided in 46 patients. An overlap of all lesion maps that were included in the analysis is shown superimposed on an axial MNI 152 template ch2better.nii.gz by MRIcron[19] in Fig. 1. The most frequent cardiovascular risk factors in the study cohort were arterial hypertension 60/85 (71%), smoking 19/85 (22%), hypercholesterolemia 18/85 (21%), diabetes 18/85 (21%) and atrial fibrillation 12/85 (14%).

### General and subdomain-specific language impairments
Aphasia score was reported for 76/85 patients (89%). In nine patients, it was not possible to obtain the ACL score due to the unavailability of the raw test data for the ACL. Median aphasia score was 135 (IQR 125–142; *n* = 76), indicating overall mild language impairment in our patient cohort. Aphasia was diagnosed in 34/76 (45%) patients. We observed a higher frequency of aphasia among those with left-sided thalamic strokes (46%) compared to right-sided strokes (33%), although this difference did not reach statistical significance (*P* = 0.274), Nonetheless, the average aphasia score was significantly lower in patients with left-sided thalamic stroke (129.0 (±15.1)) compared to patients with right-sided thalamic stroke (136.1 (±8.3)), *P* = 0.043. For a detailed comparison, please refer to Table 1 in the Supplementary material.

Ordinal scores achieved in specific subdomains were reported for all patients. Impairments were most frequently observed in the following subdomains: phonemic fluency (51/85; 60%), semantic fluency (45/85; 53%) as well as complex comprehension (42/85; 49%). Naming was less frequently impaired (15/85; 18%). Simple comprehension, reading, writing, and repeating were very rarely affected (less than five patients each). A detailed display of results in separate language subdomains with grading of impairment in each subdomain is shown in Fig. 2 and Supplementary Table 2.

### Lesion symptom mapping
We performed multivariate lesion-symptom mapping 'sparse canonical correlations for neuroimaging analyses' (SCCAN)[20] using binary, normalized lesion masks and aphasia scores to identify thalamic sites most strongly associated with worse language performance in general as well as ordinal scores of separate language subdomains for LSM subdomain-analyses. Voxels that were lesioned in less than 10% of the sample were excluded from the analysis and the lesion size was used as a co-variate.

LSM examining thalamic locations involved in more severe aphasia in general was conducted in all patients with available aphasia scores (*n* = 76). Subdomain-specific LSM analyses for each of the predefined eight language subdomains were conducted including all patients (*n* = 85).

LSM identified regions within the left ventral/anterior thalamus associated with worse language performance (aphasia score) (*r* = 0.495, *P* < 0.001, optimal sparseness = −0.242, peak coordinates: *x* = −8, *y* = −8, *z* = 4). Superimposing the resulting regions of interest (ROIs) onto Morel thalamic atlas revealed the largest overlap with the ventral part of the ventral lateral posterior nucleus (VLpv) and the left ventral anterior nucleus, parvocellular part (VApc), partly extending into the ventral lateral anterior

**Fig. 2 | Results from subdomain-specific language tasks.** Colour bars indicate number of patients with results achieved in each subdomain.

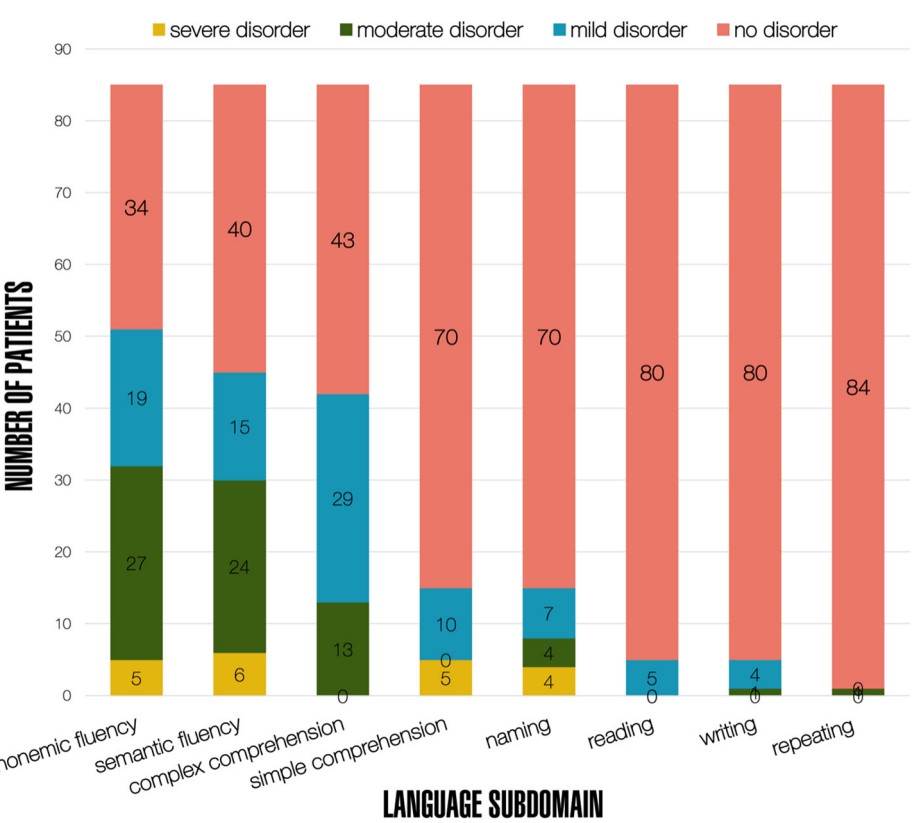

nucleus (VLa), Fig. 3, Supplementary Fig. 1. Additionally, LSM analysis revealed significant results for the subdomains *phonemic fluency* ($r = 0.482$, $P < 0.001$, optimal sparseness = 0.475, peak coordinates $x = -8$, $y = -10$, $z = 6$), *semantic fluency* ($r = 0.537$, $P < 0.001$, optimal sparseness = 0.475, peak coordinates $x = -8$, $y = -8$, $z = 4$), and *complex comprehension* ($r = 0.383$, $P < 0.001$, optimal sparseness = 0.457, peak coordinates $x = -8$, $y = -10$, $z = 6$). Lesion location for the relevant subdomains differed only slightly from the aphasia score and from each other. Superimposing ROIs of the LSM-subdomain analysis onto thalamus atlas revealed a slightly bigger overlap with the VLa (this was especially the case for *semantic fluency*) in addition to the VLpv and VApc.

No brain-behaviour relationship was found for the subdomains *simple comprehension, naming, reading, writing,* and *repeating*. Complete results of the LSM analyses for all subdomains including non-significant results are listed in Supplementary Table 3 and 4. Resulting ROIs for significant subdomains are displayed in Supplementary Fig. 1 and superimposed onto thalamic atlas in Fig. 4.

**Lesion network mapping**

The connectivity analysis utilized *connectivity profiles*, which were calculated using binary lesion masks of each subject as seeds and employing a functional normative connectome obtained from 1000 healthy individuals' resting state fMRI data[21,22]. Connectivity profiles represent voxel-wise connectivity maps based on average BOLD signal between the voxels within the lesioned area to every other voxel in the average brain. These maps were Fisher-z-transformed and used in subsequent analyses. Connectivity profiles were calculated using lead-mapper, a tool from the lead-dbs.org software[23]. To sensitively assess the strength of associations between brain regions connected to the lesion area in patients with language impairments, we first performed voxel-wise Pearson's correlations across individual connectivity profiles and aphasia scores, obtaining as a result correlation maps, and divided the subjects into two sub-cohorts to test generalizability

using cross validation in the 70% of the cohort, and predictive utility of these correlations in the remaining 30% of the cohort[24]. Only patients with available aphasia scores ($n = 76$) were considered for LNM analyses. Correlation maps of individual connectivity profiles and aphasia scores in the first sub-cohort of patients (70%, $n = 53$) showed a clear network of regions both positively and negatively connected to thalamic lesions based on aphasia scores. The correlation maps showed connectivity to both hemispheres with strongest correlation coefficients on the left hemisphere, and additional connections to the cerebellum (Crus I and II, predominantly right). *R*-values ranged from $-0.68$ to $0.71$ with FDR-corrected threshold at $-0.30$ for negative correlations and $0.30$ for positive correlations. The correlation map obtained was validated using a leave-one-out approach, obtaining an $r = 0.470$ at $P < 0.001$ (Figs. 5A and 6A). Once validated, the correlation map was used to estimate the clinical scores of the 'unseen' (30%, $n = 23$) of the cohort based on the spatial similarity of the connectivity profiles from this subset of patients to the correlation map (from the validated 70% of the cohort), obtaining an overlap with correlation coefficient $r = 0.540$ at $P = 0.008$ (Fig. 5B).

Additionally, we applied a non-parametric approach to validate the significance of the correlation maps in a more robust manner[25]. 5000 permutations were performed by using connectivity profiles as input and aphasia scores as the variable of interest to identify regions connected to lesions of patients with higher aphasia scores (worse language performance) vs. lower aphasia scores (better language performance); presence of aphasia (intended as a cut-off for clinical affection), age and lesion size were included as co-variates (Supplementary Fig. 2). The resulting maps were corrected for multiple comparison using Family-wise-error (FWE), and further corrected using threshold-free cluster enhancement (TFCE) using FSL *randomise*[26]. In the unthresholded permutation LNM analysis, bilateral (yet predominantly left-hemispheric) connectivity was observed. While lesions associated with more severe aphasia were connected to more fronto-temporal regions, thalamic lesions associated with better language performance showed

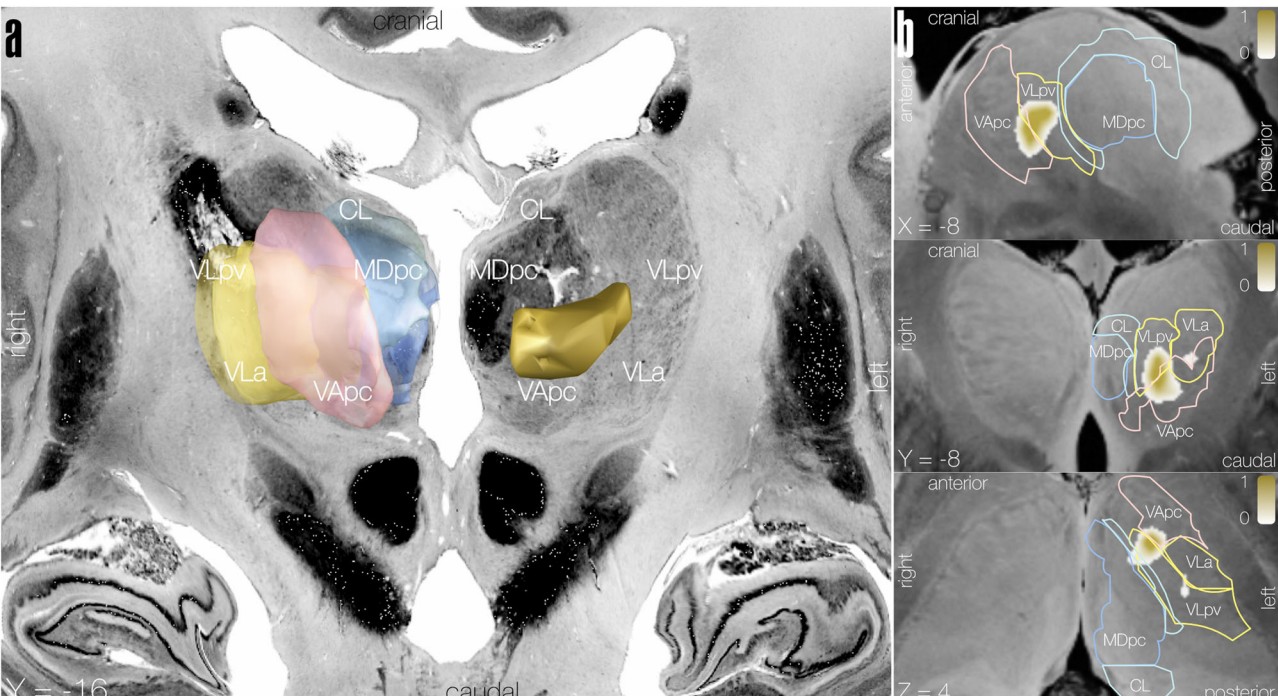

**Fig. 3 | Overlap of LSM general language impairment results with thalamic nuclei.** Resulting ROI from LSM analysis using aphasia score showing areas of overlap with thalamic nuclei using Morel thalamic atlas[53]. **a** 3D view of the ROI is shown in the left hemisphere while thalamic nuclei with the most overlap are shown magnified in their approximate location in the right hemisphere to allow visual comparison, overlaid on a brain cytoarchitecture atlas in MNI space. **b** Sagittal, coronal, and axial views of the ROI with overlapping thalamic nuclei outlines superimposed on a 100 μm resolution MNI template. CL: central lateral nucleus, MDpc: mediodorsal nucleus, parvocellular part, VApc: ventral anterior nucleus (parvocellular part), VLa: ventral lateral anterior nucleus, VLpv: ventral lateral posterior nucleus (ventral part).

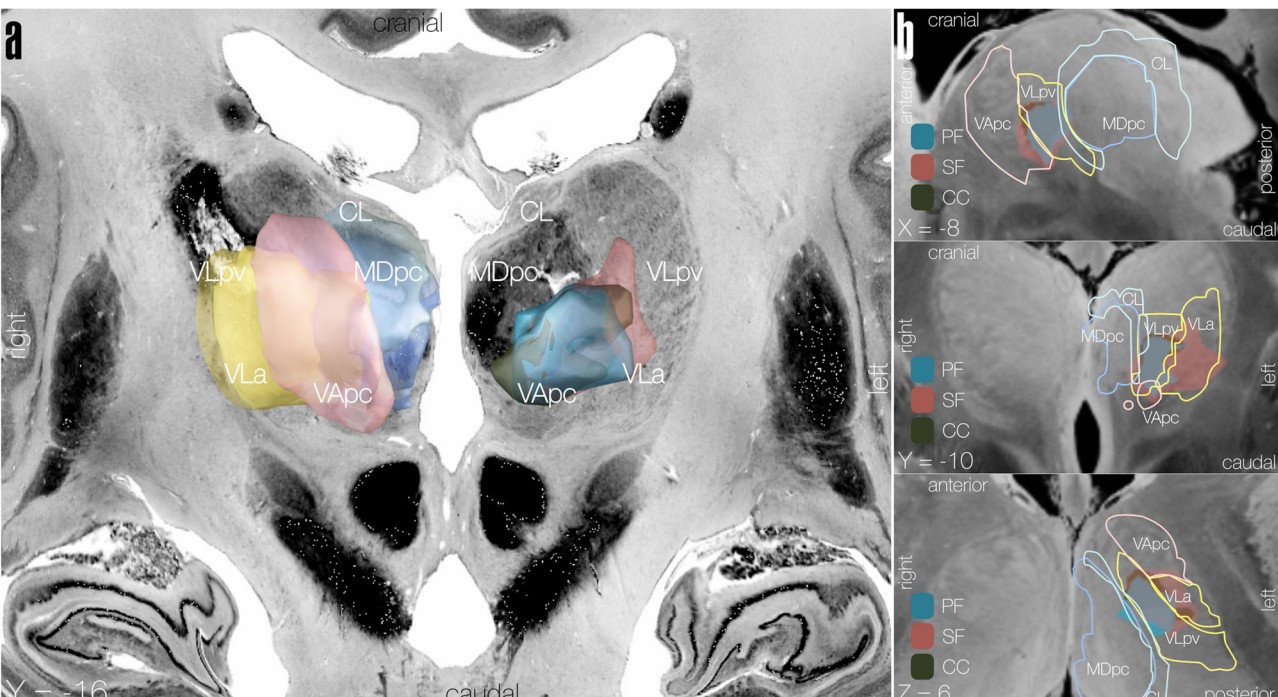

**Fig. 4 | Overlap of LSM subdomain-specific language impairment results with thalamic nuclei.** Resulting ROIs from LSM subdomain analyses showing areas of overlap with thalamic nuclei using Morel thalamic atlas[53]. **a** 3D views of subdomain-derived ROIs are shown in the left hemisphere while thalamic nuclei with the most overlap are shown magnified in their approximate location in the right hemisphere to allow visual comparison, overlaid on a brain cytoarchitecture atlas in MNI space. **b** Sagittal, coronal, and axial views of the subdomain-derived ROIs with overlapping thalamic nuclei outlines superimposed on a 100 μm resolution MNI template. Resulting ROI is shown in blue for phonemic fluency (PF), pink for semantic fluency (SF), and dark green for complex comprehension (CC). CL: central lateral nucleus, MDpc: mediodorsal nucleus, parvocellular part, VApc: ventral anterior nucleus (parvocellular part), VLa: ventral lateral anterior nucleus, VLpv: ventral lateral posterior nucleus (ventral part).

**Fig. 5 | Correlations map. a** Left: Voxel-wise map calculated from Pearson's correlation of connectivity fingerprints to aphasia scores of the 70% of the cohort (*n* = 53), thresholded at α < 0.05 FDR-corrected. Right: Leave-one-out cross-validation of correlations map to predict clinical score (x-axis), based on similarity to correlations map (y-axis, *n* = 53–1), grey shaded areas represent 95% confidence intervals. **b** Left: Pearson correlation between spatial similarity of the 'unseen' subjects connectivity profiles (30% subcohort) to the map from A (y-axis), and aphasia score prediction (x-axis), grey shaded areas represent 95% confidence intervals. Right: Voxel-wise map calculated from Pearson's correlation of connectivity profiles of the 30% of the cohort (*n* = 23), thresholded at α < 0.05, FDR-corrected.

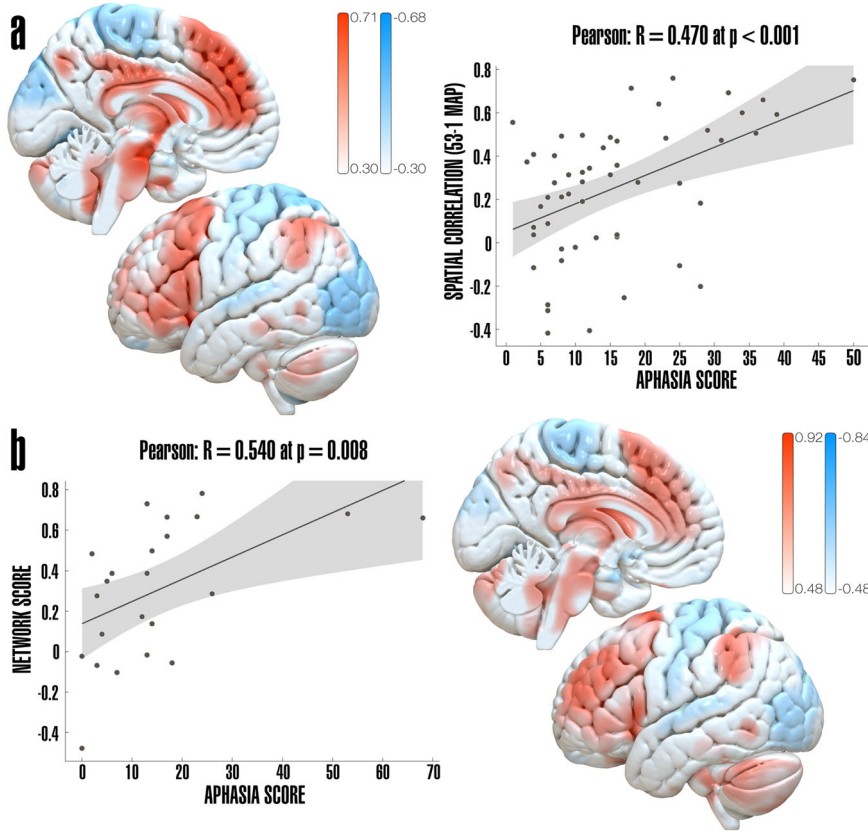

stronger connectivity to the visual cortex in both hemispheres. When thresholding to pFWE < 0.05 to isolate significant voxels, the regions showing strongest connectivity to lesions associated with more severe language impairments (from now on referred to as *Thalamic aphasia network*) included Broca's area, left middle and superior frontal gyri, precentral gyrus, cingulate and paracingulate gyri and pallidum in the left hemisphere (Fig. 6B). Relevant white matter tracts included the anterior thalamic radiation as well as the left uncinate fasciculus. Remarkably, the strongest connections to thalamic lesions causing aphasia (following threshold at significance level of pFWE < 0.02) included primarily Broca's area (inferior frontal gyrus (IFG) pars opercularis and triangularis of the left hemisphere; Fig. 7). For a complete list of regions significantly connected to thalamic lesions associated with more severe language impairments, refer to Supplementary Table 5. To ensure that the main effect observed in our results was driven by aphasia scores, an additional analysis (maintaining the model set up stable but excluding the 'presence of aphasia' variable) was performed. The resulting map remained highly similar to the Thalamic aphasia network (supplementary Fig. 3), with the addition of left caudate, middle temporal gyrus (posterior division) and cerebellar Crus I and II (right) surviving the statistical significance threshold.

To establish a comprehensive understanding of our results in the context of existing literature, we employed the decoding tool available on the Neurosynth platform (https://www.neurosynth.org)[27], which systematically compares the spatial similarity of a given network with all the maps in its database. The *Thalamic aphasia network* map resulting from lesions associated with more severe language impairment from the non-parametric approach was spatially compared to the total 1307 maps associated with terms in the Neurosynth database. Besides purely anatomical terms, such as 'prefrontal' or 'frontal gyrus', *Thalamic aphasia network* results showed the highest similarity to maps associated with the terms: 'semantic', 'retrieval', 'verb', 'word', 'language', 'sentence', 'demands', 'syntactic', 'comprehension', 'lexical' and 'phonological' (Fig. 8).

On an exploratory analysis, we investigated whether specific subdomains were responsible for more nuanced differences at the network level. The connectivity maps of the 85 patients were included as imaging data in a model that included the eight language subdomains, as specified in LSM analyses, keeping scores in an ordinal manner. Using FSL randomise, a contrast was established to extract regions connected to lesions affecting each specific subdomain and ran for 5000 permutations, using FWE as correction for multiple comparison and TFCE. The only subdomain showing statistically significant connections (pFWE 0.03–0.05) was complex comprehension, indicating the hippocampus as an associated connection to lesions affecting this subdomain.

## Discussion

In this study, we localized and characterized thalamic aphasia at the local (thalamic) and network (whole brain) level within a cohort of 85 patients who presented with (hyper)acute isolated unilateral thalamic lesions and underwent a detailed language assessment. Lesion symptom mapping (LSM) identified lesions in the left ventrolateral and ventral anterior thalamic nuclei as having the strongest association with language impairments, particularly within relevantly affected language subdomains such as verbal fluency and complex comprehension. Functional connectivity analyses using LNM revealed that thalamic lesions, which were associated with worse language performance, exhibited prominent connections to the left hemisphere, specifically involving the inferior frontal gyrus' pars triangularis and pars opercularis – most commonly recognized as Broca's area. Furthermore, we identified a widespread, predominately left-hemispheric network involved in thalamic aphasia comprising the middle and superior frontal gyri, precentral gyrus, frontal orbital cortex, cingulate (anterior) and paracingulate gyri and basal ganglia (internal pallidum) likely involving crucial white matter tracts including the anterior thalamic radiation and uncinate fasciculus.

The affected thalamic nuclei, ventral lateral (VL) and ventral anterior (VA) nuclei, belong to the thalamic motor nuclei division also known as 'effector' nuclei[6]. They are embedded in reciprocal thalamo-cortical circuits and receive input from the globus pallidus, cerebellum and motor cortex while projecting to motor, premotor and prefrontal cortical areas, including

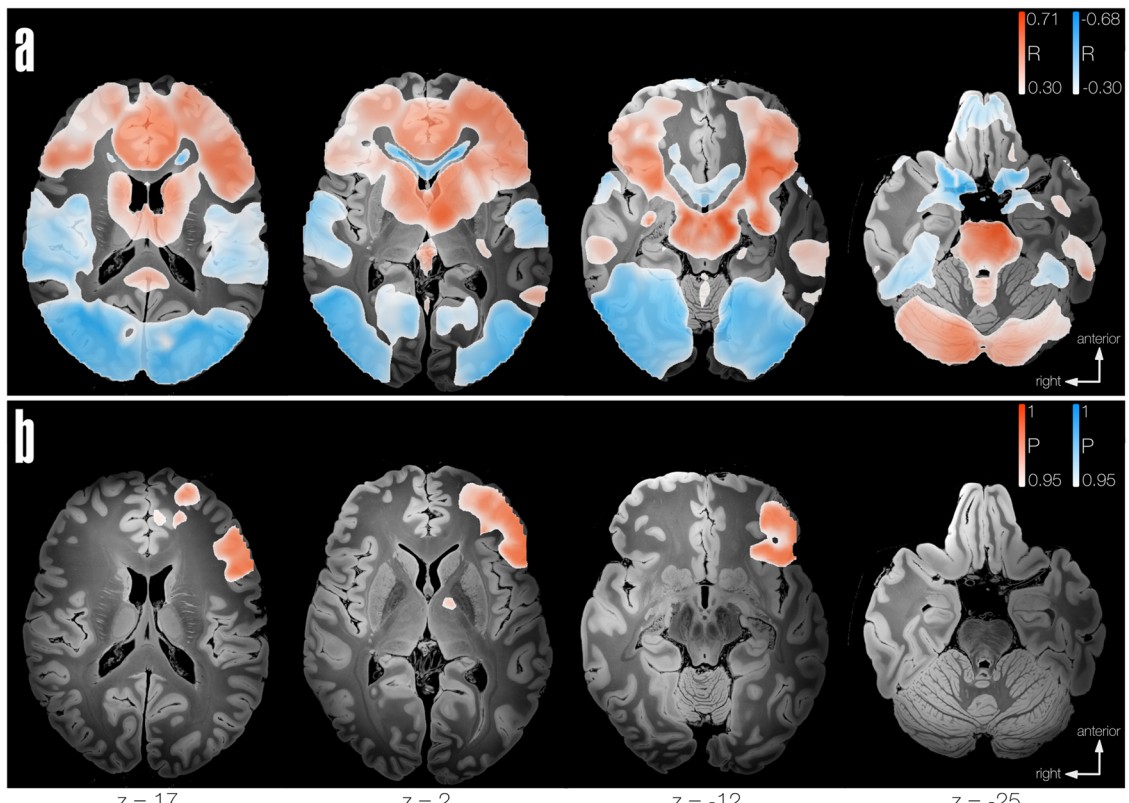

**Fig. 6 | Significant regions extracted from correlations map and thalamic aphasia network.** LNM results from **a** correlations map (70% of the cohort, *n* = 53) with a threshold at FDR α < 0.05, positive correlations (worse language performance) shown in white-red include: frontal pole, superior, middle and inferior frontal gyri, precentral gyrus, frontal orbital cortex, cingulate and paracingulate gyri, middle and inferior temporal gyri (posterior division), supramarginal (posterior division) and angular gyri, caudate, putamen and pallidum (Left > Right), brainstem, cerebellum Crus I and II (Right > Left), anterior thalamic radiation (Left > Right). Negative correlations (better language performance) shown in white-blue include: postcentral gyrus, superior parietal lobule, lateral occipital cortex, occipital fusiform gyrus and temporal (and occipital) fusiform cortex, lingual gyrus, cerebellum (VIIIb). **b** Thalamic aphasia network: FWE-corrected maps thresholded at *P* < 0.05, first contrast (worse language performance) is shown in white-red. Second contrast (better language performance) not shown since corrected map does not survive significance threshold. Also refer to Supplementary, Table 5 for complete list of significant regions.

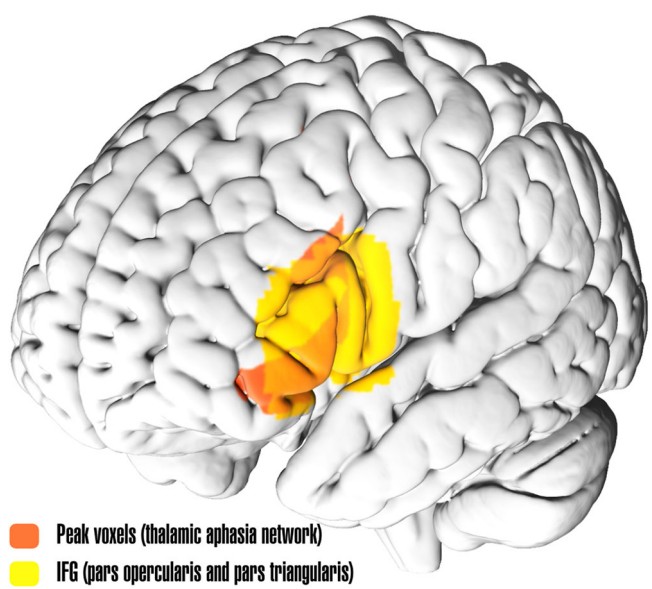

Peak voxels (thalamic aphasia network)

IFG (pars opercularis and pars triangularis)

**Fig. 7 | Strongest connections to thalamic lesions associated with worse language performance.** MNI brain surface showing the left hemisphere with superimposed peak regions from the thalamic aphasia connectivity map[62,63], thresholded at pFWE < 0.02 in red. Left inferior frontal gyrus (IFG) pars opercularis and triangularis are shown in yellow, from Harvard-Oxford Cortical Structures Atlas[55,64].

Broca's area[4]. Notably, these nuclei play a specific role in retrieval and arousal mechanisms for anterior language regions[4,28]. Supporting our results, language disturbances have been previously observed as a side effect after stereotactic surgery to the left VL for treating motor symptoms in extrapyramidal motor disorders with first evidence dating back to the 1960ies[8,29,30]. Reported symptoms included transient problems with naming, verbal memory, verbal fluency as well as deficits in Token test performance[31,32]. Two more recent studies applied LSM in thalamic stroke patients with the primary aim to identify thalamic nuclei involved in language functions. Interestingly, a study that used verbal fluency tests as a measure of language abilities also reported on involvement of the VL alongside the central medial and ventral medial nuclei[33], while another study that used retrospective chart review to evaluate language, reported involvement of the mediodorsal nucleus in patients with language impairments[34]. The variation in these results may stem from the use of diverse methods to assess language, each with varying levels of sensitivity in detecting aphasia. Additionally, a potential contributing factor to this variability could be the omission of adjustments for confounding variables, such as the impact of other cognitive domains like executive functions and verbal memory. Ultimately, it is very likely that rather than a single thalamic nucleus, there may be a hub consisting of VL, VA and mediodorsal nucleus that is essential for language processing, highlighting that in addition to thalamo-cortical connections, intra-thalamic interactions are likely as important for undisturbed language processing[4].

To assess the involvement of the thalamus in language function at a whole brain (network) level, we applied LNM using a normative human connectome. In our first LNM approach, we applied a so-called correlation

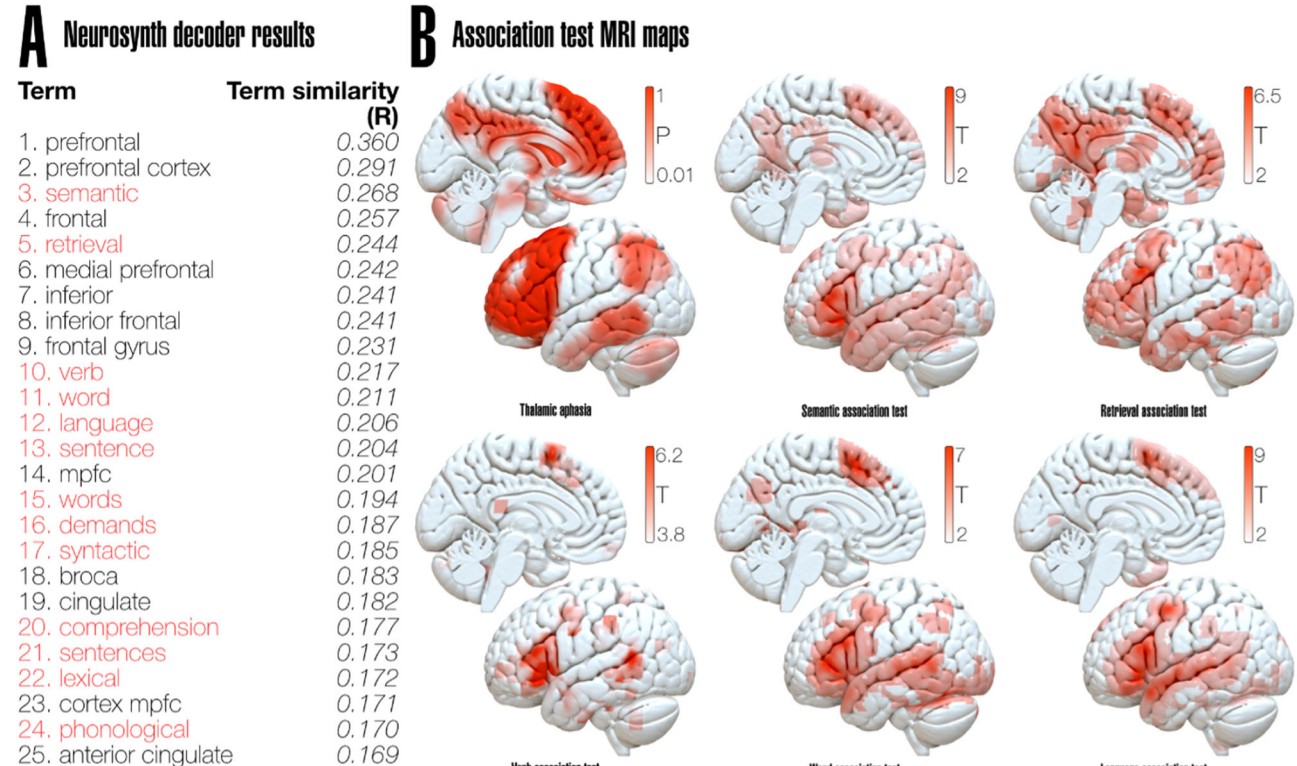

**Fig. 8 | Thalamic aphasia network compared to association maps from Neurosynth. A** Top 25 associated terms to thalamic aphasia network, purely anatomical terms are shown in black, cognitive terms related to language shown are in red. **B** Thalamic aphasia and association maps of the top five cognitive terms from Neurosynth decoder results overlaid on a brain surface.

analysis to sensitively assess the correlation of patient-specific connectivity profiles to clinical aphasia scores. Here, we not only observed that patients with thalamic lesions associated with worse language performance had highly spatially correlated connectivity profiles, but that the identified network in a subset of the cohort (70%) was able to predict clinical aphasia scores in an 'unseen' (30%) subset of patients with thalamic lesions. These results support the hypothesis that thalamic aphasia is caused by disruption of a specific network essential for normal language function. In a second LNM approach, a (rigorously constrained) non-parametric analysis was applied to identify regions most strongly connected to thalamic lesions associated with worse language performance. We observed a widespread, predominately left hemispheric network involving fronto-temporal regions in patients with worse language performance in contrast to a more dorsal network involving the parieto-occipital regions in patients with better language performance. Interestingly, lesions causing aphasia showed strongest connections to Broca's area (pFWE < 0.02), a canonical language region involved in both language-specific (language production, word retrieval, phonological and syntactic processing, among others) and domain-general cognitive functions (as a part of the so called 'multiple-demand network')[35,36], shown in Fig. 7. Moreover, we identified strong functional connectivity to left hemispheric motor areas, including the primary motor cortex, cingulate gyrus, and internal pallidum but also other prefrontal regions (superior and middle frontal gyri). Previous research underscores the pivotal role of these basal ganglia-thalamo-cortical circuits in modulating a spectrum of cognitive functions executed by the frontal regions, such as word retrieval, executive functions, attention control and associative learning processes[4,37]. These circuits resemble connections of the direct pathway, allowing initiation of motor responses through projections to the frontal cortex via the thalamus (namely VL, VA and MD) with a modulatory role coming from regions in the prefrontal cortex associated with higher-order cognitive functions, such as attention, working memory and executive function[36,38]. This is congruent with the predominantly affected language

subdomains observed in patients with thalamic aphasia – namely semantic and phonemic fluency, as well as complex comprehension which are considered higher-order language functions[39]. Nevertheless, there are noteworthy concerns related to the affected subdomains that warrant closer examination. One primary concern pertains to the influence of task selection on the conclusions drawn within our study. Notably, both verbal fluency and complex comprehension appear to significantly contribute to the interpretation of symptoms, with spatial overlap showing minimal disparities between general aphasia and subdomain-specific impairments. While we could not identify these contributions at the network level, probably due to an under representation of most subdomains, we suggest that further investigation, involving sufficiently large cohorts to enable the identification of the contributions of specific subdomains to language function is needed to allow a deeper phenotyping of thalamic aphasia. Furthermore, a secondary concern, closely related to the first one, challenges the linguistic specificity of the identified cerebral network. Both verbal fluency and complex comprehension partially depend on executive functions, with verbal fluency, in particular, being a complex task reliant on working memory, mental flexibility, and retrieval from working memory, among other cognitive functions[39–41]. Consequently, these findings may suggest potential impairments in executive functions, alongside any language-related issues. This dilemma, encountered by previous researchers in this field[42–44], including our own work[10], may warrant further comprehensive neuropsychological testing for resolution. It is essential to recognize that language operates within a broader cognitive framework. Rather than doubting the presence of aphasia in thalamic stroke altogether, the pertinent question is the nature of the specific language disorder and accompanying cognitive impairments that we are addressing.

In a systematic comparison with existing literature, we utilized the Neurosynth platform's decoding tool to compare the spatial similarity of the *Thalamic aphasia network* to other maps in their database. Remarkably, our network closely aligns with the networks associated with terms such as

"semantic", "retrieval", "verb", "word", "language", and "comprehension" (Fig. 8), providing additional validation for the association of this network with language functions. This underscores the disconnection syndrome that patients experience when the thalamic VL and VA are injured.

The advantages of this study include its prospective design, and the use of high-resolution imaging (3T MRI). Moreover, this is currently the biggest cohort of thalamic stroke patients with a detailed, standardized, homogenous, language assessment by qualified staff. Rigorous inclusion criteria were applied and only patients with acute unilateral ischemic stroke were enroled. This allowed for accurate lesion localization, segmentation, and for analysis on a voxel-wise level. Patients were included very early after symptom onset (median of 2 days), minimizing the influence of possible structural and functional reorganization processes. Combining a multivariate LSM and LNM enabled us to leverage the advantages of both methods, namely identify regions of interest at a local and functional network level. Making use of openly available, high quality, functional normative data allowed us to perform analyses on a substantial number of patients in the acute phase using only routine clinical scans without having to rely on individual functional imaging. Finally, overlaying the results of the LSM analysis onto the Morel thalamic atlas allowed us to reveal specific nuclei and even their subregions implicated in language processing which resulted in a very precise reporting of our findings and characterization of the neuroanatomy of thalamic aphasia.

Nevertheless, certain limitations apply to our study. Firstly, due to the acute setting of the study with the extensive clinical stroke work-up required during this period, we faced constraints in collecting comprehensive data. As a result, we were unable to gather information on factors such as education level and handedness. For the same reason, we did not incorporate additional neuropsychological assessment methods to account for concurrent cognitive deficits like impaired working memory, executive functions, or attention. As mentioned in the discussion, this limitation raises the question regarding whether the observed aphasic phenotype solely stems from language impairment or if it could be influenced by other cognitive functions. Addressing this conundrum would require future studies to include a broader range of cognitive assessments.

Moreover, the stroke severity in our study cohort was notably mild (median NIHSS = 2) and only patients who underwent MRI were eligible for enrolment, potentially introducing selection bias. Therefore, caution should be exercised in generalizing our findings beyond this specific population. However, our study benefits from a homogeneous cohort, which enhances the internal validity of our results and is advantageous for establishing robust lesion-behaviour relationships.

Additionally, our (hyper)acute phase, occurring within one week of symptom onset, does not account for potential neural reorganizations that may occur post-stroke. We did not conduct subanalyses based on the specific timing of examinations post-symptom onset. However, the majority of our patients (80%) were evaluated within the first three days, resulting in homogenous cohort regarding the time elapsed between symptom onset and examination. Furthermore, we were limited to demonstrating lesion-/network-language associations within thalamic regions affected in our cohort. For example, we lacked patients with pulvinar involvement, even though the pulvinar has been implicated in language processing[6,41]. Moreover, the necessary transformation into common space may lead to inaccuracies[45]. To mitigate this, each lesion mask was thoroughly reviewed to ensure the quality of transformation to standard space. Finally, our LNM analysis was limited to investigating functional connectivity from the lesions as it pertains to the average 'healthy' brain based on normative data, which prevented the consideration of potential inter-individual variations in connectivity.

In conclusion, our results make an important contribution to understanding the involvement of the thalamus in language processing on a local (thalamic) as well as on a network (whole brain) level. Due to its embedment in the frontal language circuits, we suggest that the ventrolateral and ventral anterior thalamus together with globus pallidus play a major role in programming of language and engaging the left frontal cortical areas concerned

with fluent language processing. These findings offer an important contribution to the ongoing research field on thalamic aphasia and further advance the current knowledge on the role of thalamic substructures in higher-order language tasks, taking us one step further in understanding the complexity of multimodal cerebral network involvement in language processing and ultimately closer to translating these findings into the therapeutic setting.

## Materials and methods

We prospectively enroled consecutive patients with an acute, isolated, unilateral thalamic stroke from July 2017 to August 2022 who received a detailed, standardized and validated aphasia screening test (the Aphasia Check List)[18] and 3T MRI of the brain at the stroke unit of the Department of Neurology, Charité – Universitätsmedizin Berlin, Campus Benjamin Franklin, Berlin, Germany as part of the PALAST study (Prevalence, chAracteristics und Long term course of Aphasia following Stroke in the Thalamus).

### Participants

Inclusion criteria were age ≥18 years, admission to the stroke unit at our hospital, unilateral ischemic stroke in the thalamus, symptom onset less than 7 days before study enrolment and proficiency in the German language. Exclusion criteria were bilateral thalamic lesions, hemorrhagic strokes or additional acute lesions in other cerebral arterial territories, chronic strokes in the territory of the left middle cerebral artery, pre-existing aphasia and severe neurological or psychiatric comorbidities such as depression, dementia, and substance abuse as well as uncorrected vision and substantial hearing difficulties. All patients had to be awake during the language assessment to exclude the effect of reduced wakefulness on the language assessment results.

The study was approved by the institutional review board of the Charité – Universitätsmedizin Berlin (IRB approval number EA4/199/19). Since all the data (including MRI and language assessment) were collected as part of a standard clinical work-up routine, the need for informed consent was waived by the ethics committee in accordance with local laws and regulations of the state of Berlin.

### Clinical assessment and aphasia screening

All patients underwent neurological examination on admission. Stroke severity was assessed using the National Institutes of Health Stroke Scale (NIHSS)[46]. A detailed, standardized and validated German language assessment tool, the Aphasia Check List (ACL)[18], was applied by trained board-certified speech-language pathologists (BU, MI) within 7 days after stroke onset.

A detailed description of the ACL has been published previously[18]. Briefly, ACL examines different language subdomains, including language reception and production as well as processing numbers and pseudowords. The performance in each language subdomain is rated from 0 to 3, where 3 indicates no language disorder, 2 indicates mild language disorder, 1 indicates moderate language disorder and 0 stands for severe language impairment. The subdomains were also labelled as impaired (scores 0–2) or unimpaired (score 3). We labelled the task that assesses 'everyday' auditory comprehension by carrying out easy tasks as instructed, (e.g. 'please knock on the table') as 'simple comprehension'. A colour-figure test (modification of Token Test)[47] in which patients are asked to show items with different shapes, sizes and colours with increasing complexity to assess auditory comprehension from abstract and complex verbal material in addition to verbal short term and working memory was labelled as 'complex comprehension'. Word generation tasks included *semantic* and *phonemic fluency* as well as *naming*. The subdomains reading words and sentences, pseudowords and numbers were merged into 'reading', the subdomains writing words and sentences, pseudowords and numbers were merged into 'writing' and the subdomains repeating words and sentences, pseudowords and numbers were merged into 'repeating'. For each new category, median values were calculated from the results achieved in each constituting subdomain. This resulted in eight subdomains that were tested separately in our

subdomain-specific lesion symptom mapping (LSM) analysis: *simple* and *complex comprehension* respectively, *semantic*, and *phonemic fluency* respectively, *naming, reading, writing,* and *repeating.* The total ACL score (referred to within this paper as 'aphasia score') ranges from 0 to 148, with lower scores indicating worse language performance. The aphasia score is determined using raw test data and includes not only ordinal values of subdomains ranging from 0–3 but for example also the absolute number of words said in verbal fluency tests, for details see ACL manual[18]. Aphasia is diagnosed when the aphasia score is below 135. The ACL has been validated on a normative sample using the Aachen aphasia test[48].

Using a clear cut-off definition for the presence of aphasia, ACL not only allows stratified reporting of aphasia severity in general using the continuous aphasia score, but also as a binary outcome (aphasia: yes/no) in addition to revealing the profile of affected language subdomains.

Aphasia scores were used to identify the location in the thalamus most strongly associated with worse language performance (LSM) and brain regions functionally connected to thalamic lesions associated with worse language performance (LNM). Ordinal scores were used for subdomain-specific LSM analyses.

### Neuroimaging

Patients underwent a standard stroke imaging protocol on a 3-Tesla Siemens scanner which consisted of susceptibility-weighted imaging or T2* sequence, diffusion-weighted imaging (DWI), "time-of-flight" MR angiography, and a fluid-attenuated inversion recovery sequence. DWI data set included acquisitions with a 1000 s/mm$^2$ b-value and acquisitions acquired without diffusion weighting (at 0 s/mm$^2$), as well as an apparent diffusion coefficient map. Acute thalamic strokes were diagnosed by a radiology/neurology resident and supervised by a board-certified radiologist using the b1000 DWI sequence and apparent diffusion coefficient map (TR/TE = 8900/93 ms, matrix = 192 × 192, field of view = 229 mm, slice thickness = 2.5 mm).

### MRI data pre-processing

DWI images were skull-stripped using the Brain Extraction Tool (BET) by FMRIB Software Library (FSL)[26,49]. After identifying acute diffusion restrictions on brain-extracted b1000 images, lesions were manually delineated using MRIcron (https://www.nitrc.org/projects/mricron) by a neurologist (IR) blinded to clinical aphasia outcomes. The created lesion masks were co-registered to brain extracted b0 images and in the next step normalized into common space according to Montreal Neurological Institute (MNI152 atlas, 2 × 2 × 2 mm) by performing a series of linear and non-linear registrations after masking the lesions using Advanced Normalization Tools in Python (ANTsPy)[50]. To ensure the precision of lesion segmentations and normalizations, the authors ASR and IR first visually inspected the quality of the segmentations. Following spatial normalizations of the lesions, a thorough quality check of normalized lesions was conducted by overlaying them onto the individual normalized brains to ensure accurate alignment. Subsequently, we cross-checked the alignment of the normalized thalamic lesions with the thalamic regions in the JHU atlas to validate that the normalization process did not introduce any systematic distortions. Neuroimaging methods are summarized in Fig. 9.

### Multivariate lesion symptom mapping (LSM)

A multivariate lesion symptom mapping method 'sparse canonical correlations for neuroimaging analyses' (SCCAN) was performed using the LESYMAP package available in R to identify thalamic subregions specifically involved in aphasia[20]. Binary, normalized lesion masks and aphasia scores were used as input variables for LSM analyses. SCCAN is a recently developed multivariate lesion symptom mapping method based on the assumption that a certain behaviour is unlikely to be encoded in a single voxel. Consequently, all voxels are analysed at once and weights are applied to groups of voxels that together seem to show maximal association with the

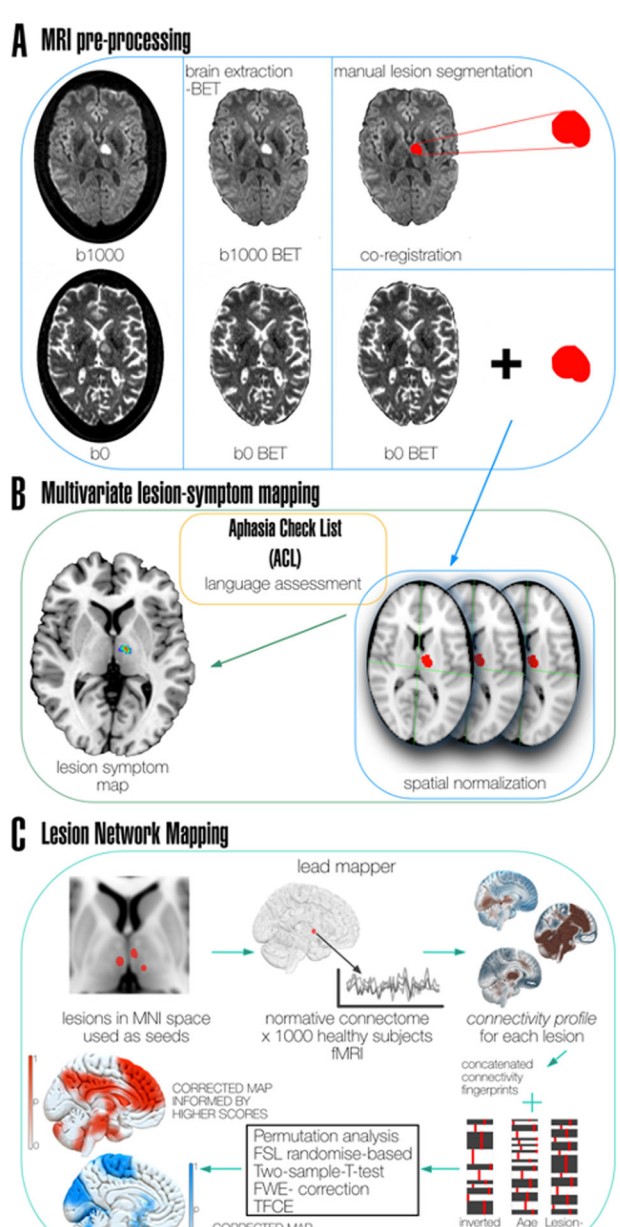

**Fig. 9 | Lesion symptom and lesion network mapping pipeline. A** MRI data pre-processing: Brain extraction on both b0 and b1000 datasets. Lesions were manually delineated on b1000 scans. Delineated lesions were co-registered with brain extracted b0 scans of each patient and standardized to MNI space using ANTs. **B** A multivariate lesion symptom mapping (LSM) method sparse canonical correlation for neuroimaging analyses (SCCAN) was performed using co-registered lesions and language assessment scores. Only voxels lesioned in at least 10% of the cases were taken into consideration, lesion size was used as a co-variate. **C** Lesion network mapping: the lesions in MNI space were used as seeds to calculate functional connectivity to the rest of the brain (based on BOLD signal) using a functional normative connectome obtained from 1000 healthy subjects' fMRI data, resulting in a connectivity profile for each lesion. Connectivity profiles were concatenated and given as input along with the clinical data (aphasia scores as variable of interest, and age and lesion size as co-variates) for a permutation analysis using a two-sample-T-test to contrast better vs. worse language performance (5000 permutations per contrast). Resulting maps were FWE-corrected and TFCE was applied. Corrected maps for each contrast in the form of 1-P were thresholded to keep only significant regions at values > 0.95.

behaviour of interest. By default, the method uses an optimization procedure that identifies the sparseness value with the best predictive accuracy by running 4-fold within-sample cross-validations at various sparseness values. At each sparseness level, a model is built using 75% of the sample. The model is then applied to the other 25% of the sample to generate predicted behavioural scores following the protocol proposed by Pustina et al.[20]. The sparseness value that achieves the highest cross-validation correlation of expected behaviour with real behaviour is selected for a final run of SCCAN with the full sample. The result of this process is a map of voxels where lesion damage is associated with the behavioural score with a unitless scale from zero to one that shows brain regions with a maximum brain-behaviour relationship, one indicating the strongest correlation. The effect size of the association between the identified brain region and the behavioural score is calculated using Pearson's correlation coefficient between the real and expected scores, including the P-value of the cross-validated correlation.

Only voxels that were lesioned in at least 10% of the cases were considered for analysis as recommended for voxel-based lesion symptom mapping analyses to avoid the risk of false positive results and to ensure sufficient statistical power by excluding rarely affected voxels[51,52]. In addition, lesion size was included as a nuisance co-variate to rule out the possibility that worse language performance was simply due to larger lesion volume[51].

First, continuous aphasia scores were used to identify thalamic sites most strongly associated with worse language performance. Second, ordinal scores of separate language subdomains were used as variables of interest for LSM subdomain-analyses to examine whether different functional regions of the thalamus play a role in different language aspects.

The resulting regions of interest (ROIs) generated using SCCAN indicating involvement with aphasia score and specific subdomains respectively were resampled to $1 \times 1 \times 1$ mm and superimposed onto Morel atlas of the human thalamus (also $1 \times 1 \times 1$ mm) using the viewer from Lead-DBS (lead-dbs.org)[23] available in MATLAB[53]. Thereby, the specific thalamic nuclei involved in more severe language impairment in general and in impairments of separate language subdomains were identified. In addition, the cluster function from Mango software (https://mangoviewer.com)[54] was used to extract the coordinates of the peak and central voxels for each derived ROI.

## Lesion network mapping (LNM)

LNM analyses were based on the methodologies proposed by Horn et al.[24] and Fox and the colleagues[12]. First, normalized binary lesion masks of separate subjects were used as seeds to calculate functional connectivity based on a normative functional connectome obtained from 1,000 healthy subjects fMRI scans[21,22], using the lead mapper tool from lead-DBS (lead-dbs.org)[23]. Voxel-wise connectivity maps were calculated between the average blood oxygenation level dependent (BOLD) signal of voxels that fell within the lesion to every other voxel in the brain of the 1000 subjects in the connectome. This resulted in a connectivity profile of each lesion mask, denoting Pearson's correlation coefficients that were averaged across 1000 normative brains. To identify the group of brain voxels most strongly connected to thalamic lesions causing more severe language deficits, connectivity profiles were Fisher-$z$-transformed and used as input for subsequent analyses.

The primary regressor for all LNM analyses were the aphasia scores. Clinical scores were kept as continuous variables to account for aphasia severity. Of note, for LNM analyses, aphasia scores were inverted (148 − individual aphasia score) so that higher scores represented worse language performance to allow a more intuitive interpretation of the data. Only patients with available aphasia scores were included in the LNM analyses ($n = 76$).

As a first statistical measure, correlation maps were created in which Pearson's correlation coefficients were calculated in a voxel-wise manner across individual connectivity profiles and aphasia scores. A corresponding R-value for each (voxel-wise) correlation coefficient was computed and thresholded at an FDR-corrected level of $\alpha = 0.05$. Since coefficients of neighbouring voxels are not independent from one another, which might result in inflated clusters[25], we randomly split the sample into two sub

cohorts: 70% ($n = 53$) and 30% ($n = 23$). First, correlation maps for the first 70% were calculated and subjected to a Leave-one-out cross-validation to evaluate generalizability. Second, correlation maps of the remaining 'unseen' 30% of the subjects were spatially correlated to the first (70%) set of maps to predict clinical scores based on their spatial similarity, evaluating the predictive utility of the identified map.

Next, as an additional statistical measure, permutations were applied using the FSL tool *randomise*[26]. Non-parametric approaches, such as this, provide robust statistical validation by making no assumptions about data distribution, allowing for correction of multiple comparison and reduction of the false positive rate[25], and thus were found suitable to authenticate the significance of the correlation maps obtained in the previous step. For this analysis, individual connectivity profiles were used as the main input, using the aphasia score as the variable of interest and presence of aphasia, age and lesion size as covariates (refer to Supplementary Fig. 2 for additional details). While traditional LNM analyses have relied on binary outcomes as independent variables, incorporating continuous variables can enhance statistical power for detecting differences among patients and increase variance. Hence, we opted to use continuous aphasia scores as the primary variable of interest. Additionally, we integrated the binary variable (aphasia yes/no) into the model to account for clinically manifest aphasia and provide more nuanced understanding of the neural network involved in thalamic aphasia. Two contrasts were established on the variable of interest (1: to identify regions connected to lesions with higher aphasia scores, i. e. with worse language performance and −1: to identify regions connected to lesions with lower aphasia scores, i. e. with better language performance) and ran for 5,000 permutations. This resulted in a 'raw' voxel-wise map for each contrast that was corrected for multiple comparison using Family-wise-error (FWE). A threshold-free cluster enhancement (TFCE) was also applied to identify cluster-forming voxels within the connected regions. The final output included FWE-corrected maps for each contrast (i) indicating worse language performance and (ii) better language performance. These maps depict 1 minus the P-value of the test association between aphasia score and the connectivity profiles. Regions were considered significantly connected to thalamic lesions if the values in the FWE-corrected maps were 1-$P > 0.95$ (corresponding to an FWE-corrected P-value (pFWE) < 0.05). An additional threshold at 1-$P > 0.98$ (pFWE < 0.02) was also applied to extract the strongest connections. This analysis was repeated excluding the 'presence of aphasia' variable from the model to confirm that the observed connections are mainly driven by lesions from patients with higher language impairment, and not by the binary cut-off for clinical aphasia.

The resulting significant voxels were overlaid onto structural atlases to identify the connected regions. The included atlases were Harvard-Oxford Cortical and Subcortical Structural Atlases[55], as well as FNIRT-normalized Cerebellar Atlas in MNI152 space[56]. Johns Hopkins University (JHU) Tractography Atlas[57] was used to evaluate connectivity to white matter streamlines.

To compare our connectivity findings with previous reports in the literature in a systematic manner, we used the decoding tool from the Neurosynth platform (https://www.neurosynth.org)[27] which compares the spatial similarity of a given network with all maps in its database and arranges the resulting spatial agreement with each term in descending order. The maps in the Neurosynth platform represent automatic meta-analyses that rely on a large number of neuroimaging studies and include 1307 terms[27,58]. For instance, the term *language* is based on 1101 studies. This tool was created to address a common misconception in fMRI research, which is to make assumptions about the actual cognitive function based solely on activation or connection sites[27,58]. This is a controversial process called reverse inference, its controversy lying in the notion that activity in most areas of the brain is non-specific across cognitive domains[45,58]. For example, in the case of Broca's area, although it is involved in language processing, it is also involved in other forms of hierarchical processing like action processing, working memory and musical processing[59–61]. To account for this pitfall and provide a basis for comparison with previous studies allowing for some reverse inference, we used the first contrast of the non-parametric results focusing on worse language performance as input for the Neurosynth decoder.

To investigate whether specific subdomains could show nuanced changes at the network level, we ran an additional exploratory permutation analysis including the ordinal scores of the eight language subdomains as variables of interest. All patients were included in this analysis (n = 85). Eight contrasts were established to extract functional connectivity to each subdomain, keeping the ordinal values of the other subdomains to investigate functional connections to lesions causing impairment of one subdomain, but not to the rest. Each contrast was run for 5000 permutations, corrected for multiple comparison with FWE and a TFCE was applied using FSL randomise, as performed for aphasia scores. A significance threshold of pFWE < 0.05 was established.

## Statistics and Reproducibility

The multivariate LSM analysis used a 4-fold within sample-cross validation by building a model with 75% of the sample at various sparseness levels and then applying it to the other 25% of the sample to generate predicted behavioural scores. The effect size of the association between the identified brain region and the behavioural score was calculated using Pearson's correlation coefficient between the real and expected scores, and the P-value of the cross-validated correlation was reported. The significance level was set at $P < 0.05$.

For the LNM analysis, Pearson's correlation coefficients were calculated voxel-wise across individual connectivity profiles and aphasia scores. Resulting correlation coefficients were thresholded at an FDR-corrected level of $\alpha = 0.05$. The sample was randomly divided into two sub cohorts: 70% (n = 53) and 30% (n = 23). Correlation maps of the first 70% were calculated and subjected to a Leave-one-out cross-validation to evaluate generalizability. Next, correlation maps of the remaining 'unseen' 30% of the subjects were spatially correlated to the first (70%) set of maps to predict clinical scores based on their spatial similarity, evaluating the predictive utility of the identified map.

In an additional permutation analysis, individual connectivity profiles were used as independent variables, aphasia scores as the variables of interest and presence of aphasia, age and lesion size as covariates. We ran 5000 permutations for two separate contrasts (regions connected to lesions associated with worse language performance vs. regions connected to lesions associated with better language performance). Resulting voxel-wise maps for each contrast were corrected for multiple comparison using FWE and TFCE was applied to identify cluster-forming voxels within the connected regions. The significance level was set at $P(FWE) < 0.05$, while $P(FWE) < 0.02$ was also applied to extract the strongest connections.

## Data availability

Behavioural data and lesion masks cannot be deposited online due to institutional regulations. However, data supporting the results of this study can be requested by qualified researchers on reasonable request by contacting the corresponding author. All requests will be thoroughly reviewed in accordance with the policies of Charité—Universitätsmedizin Berlin and the data protection laws of the state of Berlin. Supplementary data are provided with this manuscript.

## Code availability

Open source softwares were used for the pre-processing and analysis of the data, including: Lead-DBS (https://github.com/netstim/leaddbs), LESY-MAP package in R version 4.2.0 (https://github.com/dorianps/LESYMAP), FSL version 6.0.6.4 (https://fsl.fmrib.ox.ac.uk/fsl/fslwiki/) and ANTsPy (https://github.com/ANTsX/ANTsPy).

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

## Acknowledgements

We thank Dr. rer. nat. Philipp Boehm-Sturm and Dr. rer. nat. Stefan P. Koch for their valuable advice on methodological approach for lesion symptom mapping method in the early phase of this study.

## Author contributions

I.R. jointly conceived the study with C.H.N. I.R., A.S.R., A.Ku. and C.H.N. designed and performed data analysis, interpreted the results and wrote the manuscript. I.R. pre-processed the MRI data. A.H. gave conceptual and analytical advice. A.Kha., K.V. and I.G. gathered neuroimaging data and helped with image pre-processing. B.U. and M.I. performed language assessment and preprocessed the behavioural data. U.G. helped with statistical analyses. M.F., B.A.-F., and M.E., discussed the results and critically revised the manuscript. All authors read and approved final version of the manuscript.

## Funding

## Competing interests

The authors declare no competing interests.
