## [Peer review file · Communications Biology]

Reviewers' comments:

Reviewer #1 (Remarks to the Author):

General

This study investigates the role of thalamic nuclei in supporting language functions through lesion-symptom mapping in a cohort of 85 people with isolated unilateral thalamic stroke. Whether directly or through diaschisis, a specific involvement of the left ventrolateral and ventral anterior thalamic nuclei in language production is supported, because of the isolated nature of the lesions in this study, i.e., the patients crucially did not have visible structural damage outside of the thalamic region. Based on their lesion locations within the thalamus, disconnection maps were created for individual patients, which were then correlated with a general score of aphasia severity, derived from a standardized diagnostic test. The paper is well-written, and within the scope of the journal. No conceptual or technical flaws are noted, and the conclusions are largely in line with the results (see one critical note below). The incidence of isolated thalamic stroke is low, but incidence of thalamic (non-isolated) damage in stroke is much larger, so that evidence for the involvement of thalamic substructures in language functions is highly relevant both for language-theoretical reasons and for clinical management of stroke survivors who have (isolated or non-isolated) thalamic damage.

Major

- P. 10, Lesion Network Mapping: "... we first performed voxel-wise Pearson's correlations across individual connectivity profiles and aphasia scores, ..."; also p. 23: "Aphasia scores were used to identify the location in the thalamus most strongly associated with worse language performance (LSM) and brain regions functionally connected to thalamic lesions associated with worse language performance (LNM). Ordinal scores were used for subdomain-specific LSM analyses." - Why not also use the subscores for LNM analyses, perhaps in a multivariate way, to identify unique contributions of connectivity to the subdomains? Even if the associated lesions were not vastly different, is it possible that these may still be associated with subtly different connectivity profiles that might then predict differential scores on phonemic vs semantic fluency and complex comprehension, for example?
- P. 18: "In other words, connectivity findings point towards a predominately motor role of the thalamus in language functions, highlighted by the involvement of the motor cortex and internal pallidum." - I don't think that is a correct summary or conclusion; the fact that motor connections were also associated with the lesions that predict language problems does not mean these motor connections are the predominant substrate, especially since there are also non-motor-specific connections that are involved. The latter may account for the language problems, while the motor connections may simply be anatomically associated. The study does not report on any motor deficits in the patient sample, but if those exist (which is likely), one also would not conclude that Broca's area is the predominant substrate for motor movement, right?

Minor

- Demographics: Please provide data on education levels, if available, and premorbid language use: Were all patients native speakers of German?
- Demographics: It would be of interest to add data about the incidence of isolated thalamic stroke in the recruitment pool for this study, especially since reported numbers in the literature appear to vary, starting at around 3 % (Ocariz, M. d. M. S. d., Nader, J. A., Santos, J. A., &

Bautista, M. (1996). Thalamic Vascular Lesions. *Stroke*, 27(9), 1530-1536).

- P. 12: Please clarify why the 'presence of aphasia' (yes/no) was added to the model in the nonparametric validation of the correlations between aphasia scores and network characteristics - does that not reduce the variance of interest accounted for by the aphasia scores?
- It is unclear why Figure 7 depicts cortical parcellation based on the JHU atlas, while depicting the colored connection regions based on the Harvard-Oxford Cortical Structures Atlas. For clarity of illustration (and analysis), I suggest to also parcellate the brain in line with the HOCS Atlas, or to stick to labeling and depiction of results based on the JHU atlas, and to not use multiple atlases for different purposes within the same study.

Reviewer #2 (Remarks to the Author):

Rangus and Rios Infante et al. present comprehensive analyses of thalamic lesions and aphasia in a sample of 85 patients with acute ischemic stroke. By employing both lesion symptom mapping as well as lesion network mapping techniques, they find that lesions in the left ventrolateral and ventral anterior thalamic nucleus are most strongly associated with aphasia, while their thalamic aphasia network most strongly links to Broca's area and motor areas in the left hemisphere.

All in all, their analyses rely on a comparably large and well characterized dataset of patients with stroke that were recruited very soon after their stroke onset (which can be seen as advantage, as it renders any lesion symptom association more immediate). Described analyses are sound, the manuscript is well written and findings represent a relevant contribution to the understanding of thalamic involvement in language function that is currently not well characterized.

Please find some comments that are aimed at enhancing clarity and details for the authors' consideration in the following:

- It would be interesting if you could display aphasia burden separately for left and right hemispheric lesions, given that language impairment is left-lateralized in the majority of patient. Also: Did you assess handedness in your patients? One could hypothesize that mechanisms/lesion networks differ for strictly lateralized language function vs bilateral language function.
- Did you manually check the quality of lesion segmentations? As your findings are very anatomically precise, it would be an important piece of information that normalization of brain lesions did not result into any systematic distortion.
- How did you decide on the split into 70% and 30% of patients? Randomly? If it isn't rendered unfeasible due to high computational costs, it would be informative to see how this correlation holds up if you repetitively performed the 70/30 split and introduced more variability.
- Do you think that the overall mild stroke severity of patients in your sample has an effect on your results?

Point-to-point reply:

Fronto-thalamic networks and the left ventral thalamic nuclei: key players in aphasia after thalamic stroke

Ida Rangus,^{1,2,†*} Ana Sofia Rios,^{1,†} Andreas Horn,^{1,3,4,5,6} Merve Fritsch,⁷ Ahmed Khalil,^{1,2}
Kersten Villringer,^{1,2} Birgit Udke,⁸ Manuela Ihrke,⁸ Ulrike Grittner,^{9,10} Ivana Galinovic,^{1,2}
Bassam Al-Fatly,³ Matthias Endres,^{1,2,10,11,12,13} Anna Kufner,^{1,2,*} and Christian H. Nolte^{1,2,10,11,*}

Dear reviewers,

We are sincerely grateful for your invaluable comments and suggestions aimed at enhancing our manuscript. We have thoroughly examined your feedback, and we have exerted considerable effort to implement the suggested improvements to the best of our ability. Enclosed is a point-to-point response addressing each of your queries, along with the revised version of the manuscript. We trust that this point-to-point response effectively addresses your queries; however, should there be any additional feedback or unresolved concerns, we remain open and receptive to further suggestions.

Response to reviewers' comments:

Reviewer #1:

Major

• P. 10, Lesion Network Mapping: "... we first performed voxel-wise Pearson's correlations across individual connectivity profiles and aphasia scores, ..."; also p. 23: "Aphasia scores were used to identify the location in the thalamus most strongly associated with worse language performance (LSM) and brain regions functionally connected to thalamic lesions associated with worse language performance (LNM). Ordinal scores were used for subdomain specific LSM analyses."
- Why not also use the subscores for LNM analyses, perhaps in a multivariate way, to identify unique contributions of connectivity to the subdomains? Even if the associated lesions were not vastly different, is it possible that these may still be associated with subtly different connectivity profiles that might then predict differential scores on phonemic vs semantic fluency and complex comprehension, for example?

We appreciate the insightful suggestion regarding the inclusion of subdomain testing in our Lesion Network Mapping (LNM) analyses. We acknowledge the relevance of this information to our readers, particularly since we included subdomains in Lesion Symptom Mapping (LSM) analyses. In response to your suggestion, we have revisited our analyses and incorporated an exploratory LNM subdomain analysis in a multivariate manner. After examining the data, and finding borderline significant results for only one

subdomain (complex comprehension, pFWE 0.03-0.05) we find that such an analysis would require a sample size with enough power to determine the contribution of each subdomain, considering that, for instance, naming was impaired in 15 patients, and simple comprehension, reading, writing, and repeating were affected in less than 5 patients, each. For this reason, we highlight that these results should be interpreted cautiously, given that we do not have a proper representation of each subdomain to confidently extract significant contributions. The next sentences were added to the Methods and Results sections:

“To investigate whether specific subdomains could show nuanced changes at the network level, we ran an additional exploratory permutation analysis including the ordinal scores of the eight language subdomains as variables of interest. All of the patients were included in this analysis (n = 85). Eight contrasts were established to extract functional connectivity to each subdomain, keeping the ordinal values of the other subdomains to investigate functional connections to lesions causing impairment of one subdomain, but not to the rest. Each contrast was run for 5,000 permutations, corrected for multiple comparison with FWE and a TFCE was applied using FSL randomise, as performed for aphasia scores. A significance threshold of pFWE < 0.05 was established.”(Methods, p. 27 and 28)

“On an exploratory analysis, we investigated whether specific subdomains were responsible for more nuanced differences at the network level. The connectivity maps of the 85 patients were included as imaging data in a model that included the eight language subdomains, as specified in LSM analyses, keeping scores in an ordinal manner. Using FSL randomise, a contrast was established to extract regions connected to lesions affecting each specific subdomain and ran for 5,000 permutations, using FWE as correction for multiple comparison and TFCE. The only subdomain showing statistically significant connections (pFWE 0.03 - 0.05) was complex comprehension, indicating the hippocampus as an associated connection to lesions affecting this subdomain.”(Results, p. 16)

We greatly appreciate you bringing this aspect to our attention. As a response, we have enriched the discussion by highlighting the need for larger cohorts to investigate the effects of each subdomain at the network level. Our group is actively addressing this issue by actively recruiting patients within a larger cohort, ensuring sufficient power for conducting comprehensive analyses of thalamic aphasia phenotypes in the context of network-level effects. We added the following paragraph to the Discussion:

“While we could not identify these contributions at the network level, probably due to an under representation of most subdomains, we suggest that further investigation, involving sufficiently large cohorts to enable the identification of the contributions of specific subdomains to language function is needed to allow a deeper phenotyping of thalamic aphasia.” (p. 19)

- P. 18: “In other words, connectivity findings point towards a predominately motor role of the thalamus in language functions, highlighted by the involvement of the motor cortex and internal

pallidum.” - I don't think that is a correct summary or conclusion; the fact that motor connections were also associated with the lesions that predict language problems does not mean these motor connections are the predominant substrate, especially since there are also non-motor-specific connections that are involved. The latter may account for the language problems, while the motor connections may simply be anatomically associated. The study does not report on any motor deficits in the patient sample, but if those exist (which is likely), one also would not conclude that Broca's area is the predominant substrate for motor movement, right?

Upon reflection, we acknowledge that our previous conclusion may have been overly simplistic and potentially inaccurate. Our intention was to emphasize that the language functions involving the thalamus and associated networks primarily pertain to language output, as evidenced by the predominantly affected verbal fluency, a concept supported by existing literature.¹ We recognize that the mere association of motor connections with lesions predicting language problems does not necessarily imply that these motor connections represent the predominant substrate for language functions.

We appreciate your comment and believe it enhanced the interpretation of our results and the discussion. Here is the improved interpretation of our results regarding the basal ganglia-thalamo-cortical connections which can be found in the Discussion, page 18 and focusses more on specific cognitive functions involved rather than referring to 'motor role in language functions':

“Previous research underscores the pivotal role of the basal ganglia-thalamo-cortical circuits in modulating a spectrum of cognitive functions executed by the frontal regions, such as word retrieval, executive functions, attention control and associative learning processes).”

Minor

• Demographics: Please provide data on education levels, if available, and premorbid language use: Were all patients native speakers of German?

Unfortunately, we did not systematically collect data on education levels. We acknowledged this drawback in the Limitations section:

“... due to the acute setting of the study with the extensive clinical stroke work-up required during this period, we faced constraints in collecting comprehensive data. As a result, we were unable to gather information on important factors such as education level and handedness.” (p. 20)

We appreciate your attention to language proficiency in our study. As detailed in the Materials and Methods section (Participants) on page 20, while not all patients were native German speakers, it is important to note that all participants were proficient in the German language. If speech-language pathologists identified concerns regarding insufficient German language skills during initial patient screenings (which were part of the routine clinical work-up), those patients were excluded from enrollment in this study.

This approach aimed to mitigate potential confounding factors related to language proficiency and ensure the validity of our findings.

- Demographics: It would be of interest to add data about the incidence of isolated thalamic stroke in the recruitment pool for this study, especially since reported numbers in the literature appear to vary, starting at around 3 % (Ocariz, M. d. M. S. d., Nader, J. A., Santos, J. A., & Bautista, M. (1996). Thalamic Vascular Lesions. *Stroke*, 27(9), 1530-1536).

We appreciate your inquiry into the incidence of isolated thalamic stroke within our recruitment pool. It is essential to clarify, that our recruitment pool exclusively comprised patients diagnosed with isolated, MRI-confirmed thalamic ischemic strokes, enrolled consecutively following diagnosis confirmation via MRI. Consequently, we are unable to provide specific details regarding the proportion of thalamic stroke cases compared to the total stroke population treated at our stroke during the recruitment period. However, we can offer insights from a previous study conducted by our group, as outlined by Fritsch et al.² In this study, which encompassed stroke survivors admitted to our stroke unit from January 2016 to July 2017, we found that out of 1064 patients with MRI-proven acute ischemic stroke, 104 individuals (9.8%) had ischemic lesions involving the thalamus. Among these, 52 patients had an isolated ischemic lesion in the thalamus, constituting 4.9% of the initial cohort.

It's important to note that the percentage of isolated thalamic stroke cases may vary depending on factors such as the neuroimaging methods employed. For instance, thalamic infarctions may go unnoticed in the early stages of stroke if patients only undergo computer tomography. While we lack precise data for the current study cohort, we anticipate a similar percentage as observed in the Fritsch et al. study.

However, it is crucial to emphasize that determining the incidence of isolated thalamic stroke was not the objective of our present study. Instead, our focus was on establishing the anatomical correlates of thalamic aphasia. We hope this clarification adequately addresses your inquiry.

- P. 12: Please clarify why the 'presence of aphasia' (yes/no) was added to the model in the nonparametric validation of the correlations between aphasia scores and network characteristics - does that not reduce the variance of interest accounted for by the aphasia scores?

We thank the reviewer for bringing up this important point; the design of the LNM-models can vary across studies. Traditionally, binary variables were used as independent variables in LNM,³ however use of a continuous variable as the independent variable increases statistical power to detect differences among patients and increases variance. Increasingly, the method has also been applied using continuous independent variables e.g. UPDRS improvement in PD patients in DBS-network mapping studies.⁴ Therefore we decided to use the continuous aphasia scores as the variable of interest, while at the same time accounting for the clinical diagnosis of aphasia; the inclusion of the binary variable allows for an additional distinction between clinically manifest aphasia (aphasia yes/no) and provides a more nuanced understanding of the thalamic aphasia network. In other words, the model is designed to assess severity of language impairment

as well as clinically manifest 'aphasia' as a diagnosis itself. The decision to model the LNM-analysis as such was done a priori in consultation with our statistician.

We explained the rationale behind using continuous aphasia scores while also including binary aphasia outcomes (aphasia yes/no) in the Methods section, p. 26:

While traditional LNM analyses have relied on binary outcomes as independent variables, incorporating continuous variables can enhance statistical power for detecting differences among patients and increase variance. Hence, we opted to use continuous aphasia scores as the primary variable of interest. Additionally, we integrated the binary variable (aphasia yes/no) into the model to account for clinically manifest aphasia and provide more nuanced understanding of the neural network involved in thalamic aphasia.

Indeed, an alternative method would have been to assess aphasia scores (continuous) as the independent variable alone in the LNM-model. We have now also run this model on our data and provide the LNM-results in the supplementary material. Here we see a very similar network (shown in Figure 3 below this paragraph), which can serve as a further confirmatory or sensitivity analysis for the current study.

Figure 3. (Supplementary) Non-parametric analysis excluding binary outcome. Thalamic aphasia network including only aphasia scores as variable of interest, and lesion size and age as covariates. FWE-corrected map thresholded at $P < 0.05$, first contrast (worse language performance) is shown in white-red.

Additionally, we added the next sentences to the Methods and Results sections:

“This analysis was repeated excluding the ‘presence of aphasia’ variable from the model to confirm that the observed connections are mainly driven by lesions from patients with higher language impairment, and not by the binary cut-off for clinical aphasia.” (Methods, p. 27)

“To ensure that the main effect observed in our results is driven by aphasia scores, an additional analysis (maintaining the model set up stable but excluding the “presence of aphasia” variable) was performed. The resulting map remained highly similar to the Thalamic aphasia network (supplementary Fig. 3), with the addition of left caudate, middle temporal gyrus (posterior division) and cerebellar Crus I and II (right) surviving the statistical significance threshold.” (Results, p. 12)

- It is unclear why Figure 7 depicts cortical parcellation based on the JHU atlas, while depicting the colored connection regions based on the Harvard-Oxford Cortical Structures Atlas. For clarity of illustration (and analysis), I suggest to also parcellate the brain in line with the HOCS Atlas, or to stick to labeling and depiction of results based on the JHU atlas, and to not use multiple atlases for different purposes within the same study.

We thank the reviewer for highlighting this inconsistency in our figure presentation. Upon reflection, we recognize that using parcellations and labels from different atlases is a rather unusual approach. Our decision to utilize labels from the Harvard-Oxford Cortical Structures Atlas on a JHU-atlas based parcellation was made with the intention of achieving more detailed visualization provided by the JHU parcellation in comparison to a general brain surface, while remaining consistent with the atlas used to report results throughout the manuscript and on table 1 (HOOC). However, we understand the importance of maintaining consistency within one figure to avoid confusion and potential inaccuracies. In light of this, we have decided to revise our approach and utilize an MNI152 brain surface overlapping labels exclusively from the HOOC atlas for Figure 7, as shown below.

Figure 7. Strongest connections to thalamic lesions associated with worse language performance.

MNI brain surface showing the left hemisphere with superimposed peak regions from the thalamic aphasia connectivity map, thresholded at $pFWE < 0.02$ in red. Left inferior frontal gyrus (IFG) pars opercularis and triangularis are shown in yellow, from Harvard-Oxford Cortical Structures Atlas.

Reviewer #2 (Remarks to the Author):

It would be interesting if you could display aphasia burden separately for left and right hemispheric lesions, given that language impairment is left-lateralized in the majority of patient. Also: Did you

assess handedness in your patients? One could hypothesize that mechanisms/lesion networks differ for strictly lateralized language function vs bilateral language function.

Thank you bringing up this important remark. We have addressed your suggestions by adding the information on aphasia burden for the left- and right-sided thalamic strokes in the Results section:

“We observed a higher frequency of aphasia among those with left-sided thalamic strokes (46%) compared to right-sided strokes (33%), although this difference did not reach statistical significance ($p = 0.274$), Nonetheless, the average aphasia score was significantly lower in patients with left-sided thalamic stroke ($129.0 (\pm 15.1)$) compared to patients with right-sided thalamic stroke ($136.1 (\pm 8.3)$), $p = 0.043$. For detailed comparison, please refer to Table 1 in the Supplementary material.” (Results/General and subdomain-specific language impairments, p. 6)

Additionally, we added a table showing results of the presence of aphasia and aphasia scores separately for the left and the right side in Table 1 in the Supplementary material.

Regrettably, we did not assess handedness due to the acute nature of the study, which limited our ability to gather comprehensive data from patients. This limitation is duly acknowledged in the limitations section of our study.

“... due to the acute setting of the study with the extensive clinical stroke work-up required during this period, we faced constraints in collecting comprehensive data. As a result, we were unable to gather information on important factors such as education level and handedness.” (Discussion/Strengths and limitations, p. 20)

Did you manually check the quality of lesion segmentations? As your findings are very anatomically precise, it would be an important piece of information that normalization of brain lesions did not result into any systematic distortion.

We appreciate your inquiry regarding the quality assurance measures of lesion segmentations and spatial normalization in our study. To ensure accuracy, the quality of lesion segmentations was visually inspected by ASRI and IR. Additionally, we conducted a thorough examination of spatial normalization by overlaying the normalized lesions onto the normalized brains of individual participants. We subsequently checked alignment with the JHU atlas to verify that there were no systematic distortions introduced during the normalization process.

We recognize the importance of transparently showcasing this methodological step to our readers. Therefore, we included a detailed description of these quality control procedures in the Methods section of our manuscript. This will provide clarity and assurance regarding the reliability of our findings:

“To ensure the precision of lesion segmentations and normalizations, the authors ASR and IR first visually inspected the quality of the segmentations. Following spatial normalizations of the lesions, we conducted a thorough quality check of normalized lesions

by overlaying them onto the individual normalized brains to ensure accurate alignment. Subsequently, we cross-checked the alignment of the normalized thalamic lesions with the thalamic regions in the JHU atlas to validate that the normalization process did not introduce any systemic distortions.” (Methods, p. 24)

How did you decide on the split into 70% and 30% of patients? Randomly? If it isn't rendered unfeasible due to high computational costs, it would be informative to see how this correlation holds up if you repetitively performed the 70/30 split and introduced more variability.

Thank you for your inquiry regarding our methodology for splitting the patient cohort into 70% and 30% subsets for our analysis. The decision to split the data in this manner was made a priori following consultation with our statistician (UG). This split ratio is commonly used in predictive modeling and machine learning tasks, aiming to strike a balance between training and testing datasets while ensuring sufficient data for validation. However, we understand the importance of demonstrating the robustness of our results through multiple splits.

To address this concern, we have considered employing more advanced techniques, such as repetitive splitting of the data or utilizing machine learning approaches. While these methods offer potential benefits, they also introduce complexities and computational costs that are beyond the scope of our current analysis.

Nevertheless, we have taken steps to ensure the robustness of our results. We have followed the suggestions by Poldrack and colleagues, which include non-parametric testing with multiple test correction to reduce inflated false positive rates in our analyses.⁵ This approach provides additional confidence in the reliability of our findings within the constraints of our study design. We appreciate your suggestion and the opportunity to clarify our methodology. Should future studies permit, we will explore more advanced approaches to further strengthen the robustness of our analyses.

Do you think that the overall mild stroke severity of patients in your sample has an effect on your results?

We investigated a highly selective sample of acute stroke patients with isolated infarct in the thalamus. As highlighted by the reviewer, our study population exhibited relatively mild stroke severity, evident from the median NIHSS score of 2, a well-documented observation in thalamic stroke cases. To ensure thorough assessment and precise characterization of aphasia in this context, we employed a highly sensitive language assessment – the Aphasia Check List. This approach enabled us to overcome the limitations associated with less detailed language assessments, such as neurological exam and NIHSS, which would not have allowed to study aphasia in our sample in depth as we did.

It is also important to recognize that our study's reliance on MRI for patient inclusion may introduce selection bias towards individuals with milder strokes who are more likely to undergo MR-neuroimaging. This could potentially limit the applicability of our findings to a broader stroke population, although we would like to stress that the aim of our study was to examine aphasia in thalamic stroke patients specifically and not in a broader population of stroke survivors. By addressing these points, we aim to provide a

comprehensive understanding of how the mild stroke severity in our sample may influence the interpretation and generalizability of our results:

“Moreover, the stroke severity in our study cohort was notably mild (median NIHSS = 2) and only patients who underwent MRI were eligible for enrollment, potentially introducing selection bias. Therefore, caution should be exercised in generalizing our findings beyond this specific population. However, our study benefits from a homogeneous cohort, which enhances the internal validity of our results and is advantageous for establishing robust lesion-behaviour relationships.” (Discussion/Strengths and limitations, p. 20)

Kind regards,

Ida Rangus, MD
on behalf of all co-authors

REVIEWERS' COMMENTS:

Reviewer #2 (Remarks to the Author):

I would like to thank the authors for their comprehensive revision that has satisfactorily addressed my concerns.

As a final minor comment: I would suggest to add in your methods section that you randomly split the cohort into 70% and 30% (and not e.g. based on specific selection criteria deciding upon who is in the 70 or 30% group), thank you.